# Comprehensive analysis of molecular characteristic and clinical prognosis of CD8+ T cell related genes in idiopathic pulmonary fibrosis

Lin Feng[1,2], Jiabo Yuan[3], Xiaobing Dou[1]*, Yanlin Liu 🄐 [1,4]*

1 Life Sciences, Zhejiang Chinese Medical University, Hangzhou, Zhejiang, China, 2 Hunan University of Medicine, Huaihua, Hunan, China, 3 Heilongjiang University of Chinese Medicine, Harbin, Heilongjiang, China, 4 Acupuncture and moxibustion Department, Zhejiang Chinese Medical University Affiliated Jiaxing TCM Hospital, Jiaxing, Zhejiang, China

* xbdou77@163.com (XD), liuyanlin2193@163.com (YL)

## Abstract

Idiopathic pulmonary fibrosis (IPF) is a progressive, life-threatening interstitial lung disease whose pathogenesis remains unclear. There is evidence showing the possible role of CD8+ T cells in the pathogenesis of IPF and the correlation with the clinical symptoms of IPF. In order to further explore the role of CD8+ T cells in IPF, we screened CD8+ T cell related genes (CRG) that are associated with IPF prognosis, and established the molecular typing characteristics of IPF. Subsequently, CXCR4, GPR56 and PAK1 were screened as independent prognostic factors. Expression profiles and multivariate analysis coefficients were used to establish and validate prognostic features of IPF. Immuno-infiltration characteristics of the established feature were also analyzed. Subsequent *in vitro* experiments verified the abnormal expressions of three independent prognostic factors in TGF-β1 treated IPF model at protein and mRNA levels. Our findings shed new light on the important role of CD8+ T cells in the pathogenesis of IPF and provide potential targets for predicting prognosis and possible future clinical applications.

## Introduction

Idiopathic pulmonary fibrosis (IPF) is a progressive, life-threatening interstitial lung disease with unknown pathogenesis [1]. The clinical manifestations of IPF are progressive dyspnea and significantly reduced lung compliance [2]. Due to secondary respiratory failure, patients with IPF usually die within 5 years of diagnosis [3]. Although the cause of IPF is unknown, there is substantial evidence for the potential role of T cells in the progression of fibrosis. [4,5]. Therefore, further understanding of the role of immune cells in the alveolar microenvironment is necessary for the complex crosstalk of extracellular matrix (ECM) and other cellular components.

**Data availability statement:** All relevant data are within the manuscript and its Supporting information files.

**Funding:** This work was supported by Zhejiang Province Traditional Chinese Medicine Science and Technology Plan Project: 2025ZL568 to Y.L.; Scientific Research Project of the Affiliated Hospital of Zhejiang University of Traditional Chinese Medicine: 2023FSYYZQ15 to Y.L.; Key Laboratory of Rehabilitation Study of Integrative Traditional Chinese and Western Medicine in Jiaxing City: Jiakegao 2022[38] to Y.L.; Jiaxing City Science and Technology Bureau project: 2024AD30075 to Y.L.

**Competing interests:** The authors have declared that no competing interests exist.

Numerous studies have shown that immune cells play a role in IPF [6]. Many IPF patients have unexplained elevation of autoantibodies, and some autoantibodies are associated with acute exacerbation of IPF [7,8]. Analysis of fibrotic samples from IPF patients showed activation of T-cell co-stimulatory genes and downregulation of inhibitory immune checkpoint genes, suggesting activation of immune responses in the samples. At the same time, the samples showed increased infiltration of CD8[+] T cells [9]. Analysis of extracellular vesicles similarly showed that IPF patients had significantly higher numbers of CD8[+] T cells than controls [10]. Studies on the role of CD8[+] T cells in the pathological process of fibrosis have shown that the response of CD8[+] T cells to viruses is an important contributor in the process of fibrosis formation in mouse fibrosis models. Depletion of CD8[+] T cells helped protect mice from fibrotic disease [11]. In addition, the key role of CD8[+] T cells in the induction of splenic fibrosis also suggests its influence on the pathological process of fibrosis [12].

Including CD8[+] T cells, blood immune phenotype closely related with IPF disease severity and progress [13]. In particular, single-cell RNA sequencing of peripheral blood mononuclear cells (PBMCs) from IPF patients revealed a significant increase in effector CD8[+] T cells, which may reflect disease activity and immune dysregulation [14]. In addition to the higher degree of infiltration in patients with IPF, the association between CD8[+] T cells and the severity of clinical symptoms in patients with IPF suggests their potential as a prognostic target. Results of surgical biopsies of lung tissue from IPF patients showed that CD8[+] T cells were associated with functional parameters of grade of dyspnea and disease severity [15]. Lymphocyte subsets in bronchoalveolar lavage fluid can reflect the pattern of lymphocyte infiltration in lung parenchyma [16]. It has been reported that CD8[+] T cells in bronchoalveolar lavage fluid are associated with the degree of dyspnea caused by IPF [17]. Therefore, further study of CD8[+] T cell-related pathways and related proteins may help delay the pathological process and improve the accuracy of prognosis prediction of IPF.

In this study, we screened CD8[+] T cell related genes (CRG) associated with IPF prognosis and established molecular typing characteristics of IPF. Subsequently, the expression profiles and multivariate analysis coefficients of three independent prognostic factors (CXCR4, GPR56 and PAK1) were used to establish and verify the prognostic characteristics of IPF. After analyzing the immunoinfiltration characteristics of CRG, we re-verified the abnormal expression of independent prognostic factors in transforming growth factor-β1 (TGF-β1) treated IPF model through *in vitro* experiments. Our results show the important role of CD8[+] T cells in the pathogenesis of IPF from a new perspective, and provide potential targets for predicting prognosis and possible future clinical applications.

## Materials and methods

### IPF dataset collection

In this study, we retrieved transcriptomic and clinical data for IPF patients and healthy donors from the GSE70866 dataset, which is based on gene expression profiling of bronchoalveolar lavage (BAL) cells. This dataset contains samples generated from two different microarray platforms. After filtering out duplicate entries and retaining

only samples with complete clinical information, a total of 176 IPF patients and 20 healthy donors were included in the subsequent analysis. To mitigate potential batch effects introduced by the use of different platforms, we used the "SVA" package in R to identify and remove surrogate sources of variation, followed by normalization with the "limma" package. Corresponding clinical parameters, including survival time, survival status, age, gender, and GAP index, were extracted from the sample annotation files (Supplementary table 1).

### Identification of key gene modules in immune cells and differential analysis

To determine the most critical gene module associated with immune cells in IPF, we utilized the "WGCNA" R script (Weighted Gene Co-expression Network Analysis) to calculate the correlation between gene modules and immune cells. Initially, we performed cluster analysis on all samples and subsequently excluded any outlier samples. Following the removal of outliers, we selected the optimal soft threshold to establish a scale-free network, utilizing dynamic tree cutting to integrate the acquired gene modules. For the gene modules obtained, Pearson correlation analysis was utilized to calculate the correlation coefficient between 23 immune cells and different gene modules, with the most significant gene module chosen for subsequent analysis (Supplementary table 2). Under the condition of a threshold with $p$ (adjust) < 0.05, we employed the "limma" R script to perform a differential expression analysis of the healthy donors and IPF groups.

### Establishment and verification of CRG prognostic signature for IPF

The "venn" script extracted common genes between differentially expressed genes (DEGs) and CRG to identify potential prognostic CRG. We performed univariate Cox analysis using the "survminer" script to assess the hazard ratio (HR) and P-value of DE-CRG, considering adjust. $p$-value < 0.05 as an indicator of prognostic CRG. Then, we conducted Least Absolute Shrinkage and Selection Operator (LASSO) analysis to examine prognostic variables and multivariate Cox analysis to identify independent prognostic variables. We established the CRG score of each IPF sample based on the expression and coefficient values of CXCR4, GPR56, and PAK1. Subsequently, IPF samples were split into low- and high-CRG score subgroups based on the median CRG score. To ensure the robustness and generalizability of the CRG scoring model, we used the "caret" R package to randomly split all IPF samples with complete survival information into a training set and a validation set at a ratio of 7:3. The training set (n = 124) was used for model construction, including feature gene selection and risk score calculation, while the validation set (n = 52) was used to independently evaluate the predictive performance of the model. The "survival" script then estimated the clinical outcome of IPF in the CRG score subgroups. We also produced time-related Receiver Operating Characteristic (ROC) curves and calculated the area under the curve (AUC) for 1-, 3-, and 5-year survival times using the "survivalROC" package.

### Unsupervised consensus clustering analysis to identify molecular subtype characteristics

To investigate IPF molecular subtypes based on independent prognostic variables, we employed unsupervised consensus clustering analysis using the "ConsensusClusterPlus" script. We set the range of k between 2–9 and identified the optimal number of clusters, evaluating the quality of clustering based on consensus values, cumulative distribution function (CDF), and proportion of ambiguous clustering (PAC). Consequently, we divided IPF samples into distinct CRG-based subtypes, and we evaluated the clinical outcome of each subtype using the "survival" script. Lastly, we utilized the "ggalluvial" script to explore the possible association between clinical survival prognosis, CRG score, and CRG molecular subtypes.

### Independence assessment and nomogram construction

We carried out both univariate and multivariate Cox regression analyses to evaluate the independent prognostic value of CRG score, age, gap and gender in IPF. Through the integration of these variables, we determined the independence of each variable and its impact on clinical outcomes. Also, using the "rms" script, we developed a nomogram to help assess

the 1-, 3-, and 5-year survival probability of IPF patients based on their individual characteristics. We employed the "pROC" package to examine the diagnostic power of CRG score, gap, age, and gender.

## Cell culture and treatment

Human embryonic lung fibroblasts (MRC-5) were purchased from Cell Resource Center (Chinese Academy of Medical Sciences, Beijing, China) and cultured in MEM/EBSS medium containing 10% FBS, $1 \times 10^5$ U/L penicillin and 100 g/L streptomycin at 37°C and 5% CO2. The MRC-5 cells induced by 3 µg/L TGF-β1 were used as a pulmonary fibrosis cell model. TGF-β1 was purchased from Sigma-Aldrich (St. Louis, MO, USA; Cat. 616455).

## Western blot

Cell lysates were prepared using RIPA buffer, followed by separation of proteins by SDS-PAGE and transfer to a PVDF membrane. Subsequently, the membrane was incubated with specific primary antibodies followed by secondary antibodies and then observed using the Odyssey Clx imaging system (Lincoln, NE, USA). The specific primary antibodies used in this study, including anti-CXCR4 (Cat. ab181020), anti-GPR56 (Cat. ab302909), and anti-PAK1 (Cat. ab223849), were purchased from Abcam (Cambridge, UK), and anti-β-actin (Cat. 4970) antibody was obtained from Cell Signaling Technology (CST, Danvers, MA, USA). Band intensities were quantified using Quantity One V 4.62 software (Bio-Rad, Hercules, CA, USA).

## qRT- PCR analysis

Total RNA was extracted from cell samples using the TRIzol reagent (Invitrogen, Thermo Fisher Scientific, Waltham, MA, USA; Cat. 15596026) according to the manufacturer's instructions. Complementary DNA (cDNA) was synthesized using the PrimeScript™ RT reagent kit (Takara Bio Inc., Shiga, Japan; Cat. RR047A). Quantitative real-time PCR (qRT-PCR) was performed in triplicate for each sample using the Mx3000P Real-Time PCR System (Agilent Technologies, Santa Clara, CA, USA). The β-actin gene was used as an internal control to normalize mRNA expression levels. Relative quantification was calculated using the method described by Xue et al. [18].

## Statistical analysis

All statistical analyses were performed using R software (version 4.4.1), GraphPad Prism (version 8.0.1), and SPSS 18.0 (SPSS Inc., Chicago, IL, USA). All *in vitro* experiments, including quantitative real-time PCR and Western blot, were independently repeated at least three times to ensure reproducibility. Data are presented as the mean ± standard deviation (SD). For comparisons between two groups, either the unpaired two-tailed Student's t-test and the Wilcoxon rank-sum test was applied. For comparisons among multiple groups, one-way analysis of variance (ANOVA) test was conducted. To account for multiple comparisons, P-values were adjusted by FDR (false discovery rate) method accordingly. A adjust. *p*-value of less than 0.05 was considered statistically significant, with significance levels indicated as *$p < 0.05$, **$p < 0.01$, and ***$p < 0.001$.

## Results

### Identification of genes modules most relevant to immune cells in IPF

In this study, we obtained a total of 196 samples (20 healthy donors and 176 IPF samples) from the GSE70866 dataset to investigate the potential role of immune cell related genes in IPF. Utilizing the ssGSEA algorithm, we assessed the proportions of 23 immune cells in both normal and IPF samples. By setting a soft threshold of 12, we constructed a free-scale network (Fig 1A) and derived 13 different gene modules for subsequent analysis, by cutting and merging the obtained gene modules using the dynamic tree method (Fig 1B). The heatmap of module-trait correlations revealed significant

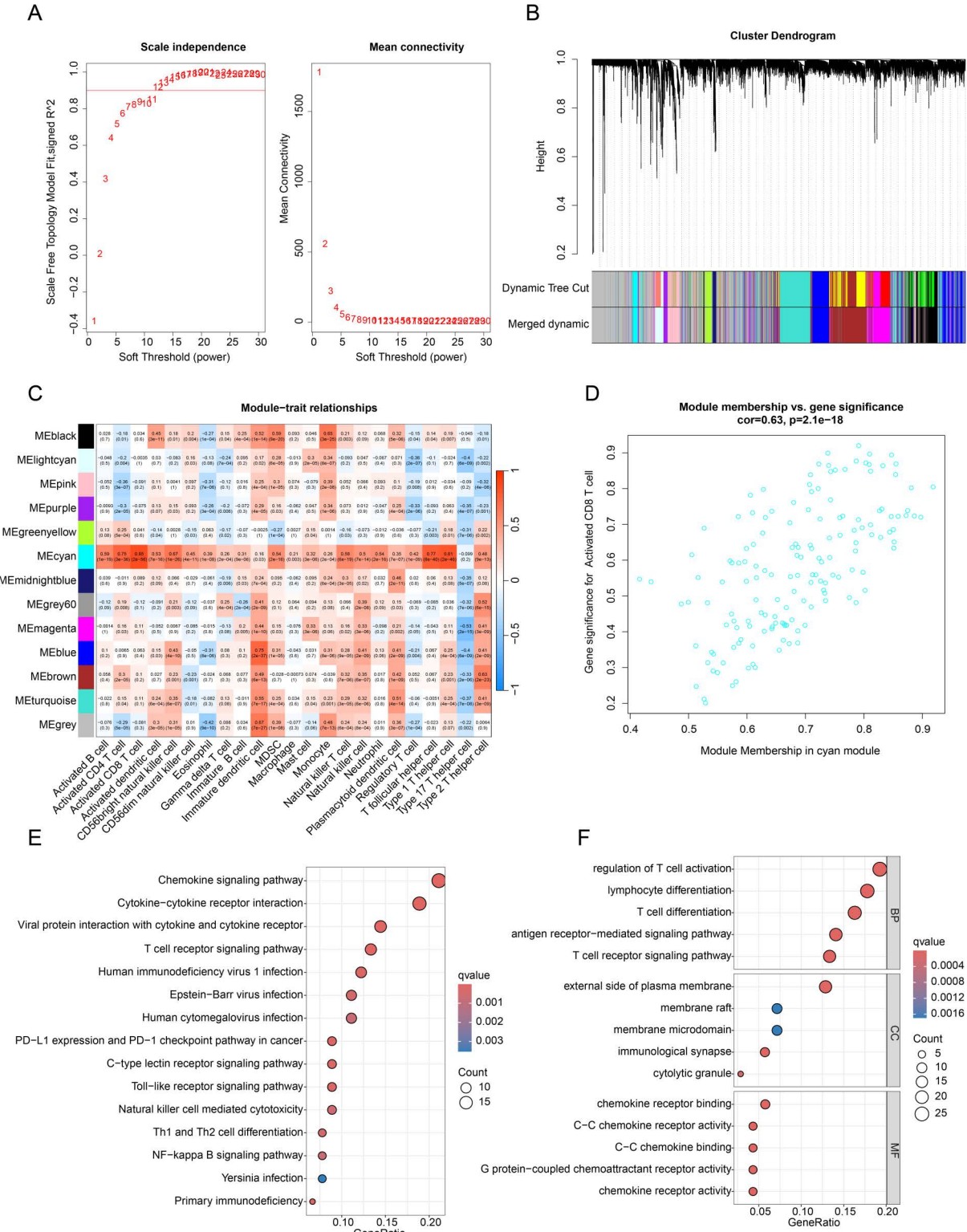

**Fig 1. Identification of key immune cell-related gene modules through the WGCNA algorithm. (A)** Selection of the most appropriate soft threshold to construct the WGCNA network. **(B)** Cutting and merging gene modules using the dynamic tree method. **(C)** Heatmap of module-trait correlations between gene modules and 23 immune cell features. **(D)** Scatter plot showing the correlation between the cyan gene module and CD8+ T cells.

associations between different gene modules and the 23 immune cells (Fig 1C). Notably, we observed a significant positive correlation between the cyan module and the majority of immune cells, especially CD8$^+$ T cells (r = 0.85, $p$ = 2e-56). The scatter plot also demonstrated a significant positive correlation between membership of the cyan module and gene significance of CD8$^+$ T cell-related genes (r = 0.63, $p$ = 2.1e-18, Fig 1D). The results of KEGG enrichment analysis indicated that the module genes were primarily associated with immune-related signaling pathways, including the Chemokine signaling pathway, Cytokine–cytokine receptor interaction, T cell receptor signaling pathway, and Viral protein interaction with cytokine and cytokine receptor. GO enrichment analysis further revealed that these genes were involved in regulating key cellular functions such as T cell activation, T cell differentiation, T cell receptor signaling pathway, and chemokine receptor binding (Fig 1E, 1F). Therefore, we postulate that CD8$^+$ T cells might be the most relevant immune cells in regulating IPF and we collated the genes in the cyan module for further analysis.

## Identification of differentially expressed CRG and prognosis value evaluation in IPF

In order to investigate the potential role of CRG in IPF, we utilized the "limma" R package to conduct an analysis of differentially expressed CRG between healthy donors and IPF groups. Our analysis revealed 5,073 genes with differential expression between these two groups at a threshold of $adjust.p < 0.05$ (Fig 2A). Furthermore, using WGCNA and differential analysis, we identified 48 differentially expressed CRG using a Venn diagram (Fig 2B). To assess the potential prognostic value of these CRG in IPF, we performed univariate Cox analysis, incorporating clinical survival data of IPF samples. Notably, our network diagram results revealed a significant positive correlation between the 48 differentially expressed CRG, and we identified 21 risk factors related to IPF prognosis (Fig 2C, HR > 1, $p < 0.05$). Additionally, we performed LASSO analysis to further screen the feature factors, and we found that the log lambda value was smallest when the variable was 7 (Fig 2D). Lastly, results from our multivariate Cox analysis suggested that three CRG, namely CXCR4, GPR56, and PAK1, may serve as independent prognostic factors for IPF.

## The molecular subtyping characteristics of IPF

We employed unsupervised consensus clustering analysis to explore the molecular subtypes of IPF based on the expression profiles of three independent prognostic factors. The consensus clustering model was most reliable when k = 2, as IPF samples were grouped into two CRG molecular subgroups (Fig 3A). The clinical prognostic survival outcome curves revealed that the survival prognosis of IPF samples in CRG subgroup A was significantly worse than that of samples in subgroup B (Fig 3B). The PCA plot displayed two significantly separated distribution patterns of CRG subgroups A and B (Fig 3C). Immune infiltration analysis indicated that the proportion of most immune cells in CRG subgroup A with poor prognosis was higher compared to CRG subgroup B. These immune cells included activated B cells, CD4 + T cells, CD8$^+$ T cells, activated dendritic cells, and CD56bright natural killer cells. These results suggest that the immune status of IPF samples in CRG subgroup A may be higher (Fig 3D). To gain a better understanding of the potential mechanisms between molecular subtypes of IPF, we analyzed differentially expressed genes between the two subtypes using the "limma" script. The GO enrichment analysis revealed significant enrichment of DEGs between the two subgroups in leukocyte migration, external side of plasma membrane, cytokine receptor binding and cell chemotaxis related pathways (Fig 3E). The KEGG results indicated that some immune-related signaling pathways were significantly enriched, such as cytokine-cytokine receptor interaction, chemokine signaling pathway, and viral protein interaction with cytokine and cytokine receptor (Fig 3F). Based on these results, we speculate that immune-related signaling pathways may be a key mechanism regulating CRGs subtypes, and may be associated with poor clinical prognosis.

## Construction of a prognostic model based on CRG prognostic variables

Using the expression profile of three independent prognostic factors and multivariate analysis coefficients, we assessed the risk score of each IPF sample and developed a novel prognostic risk model, represented by the CRG score formula:

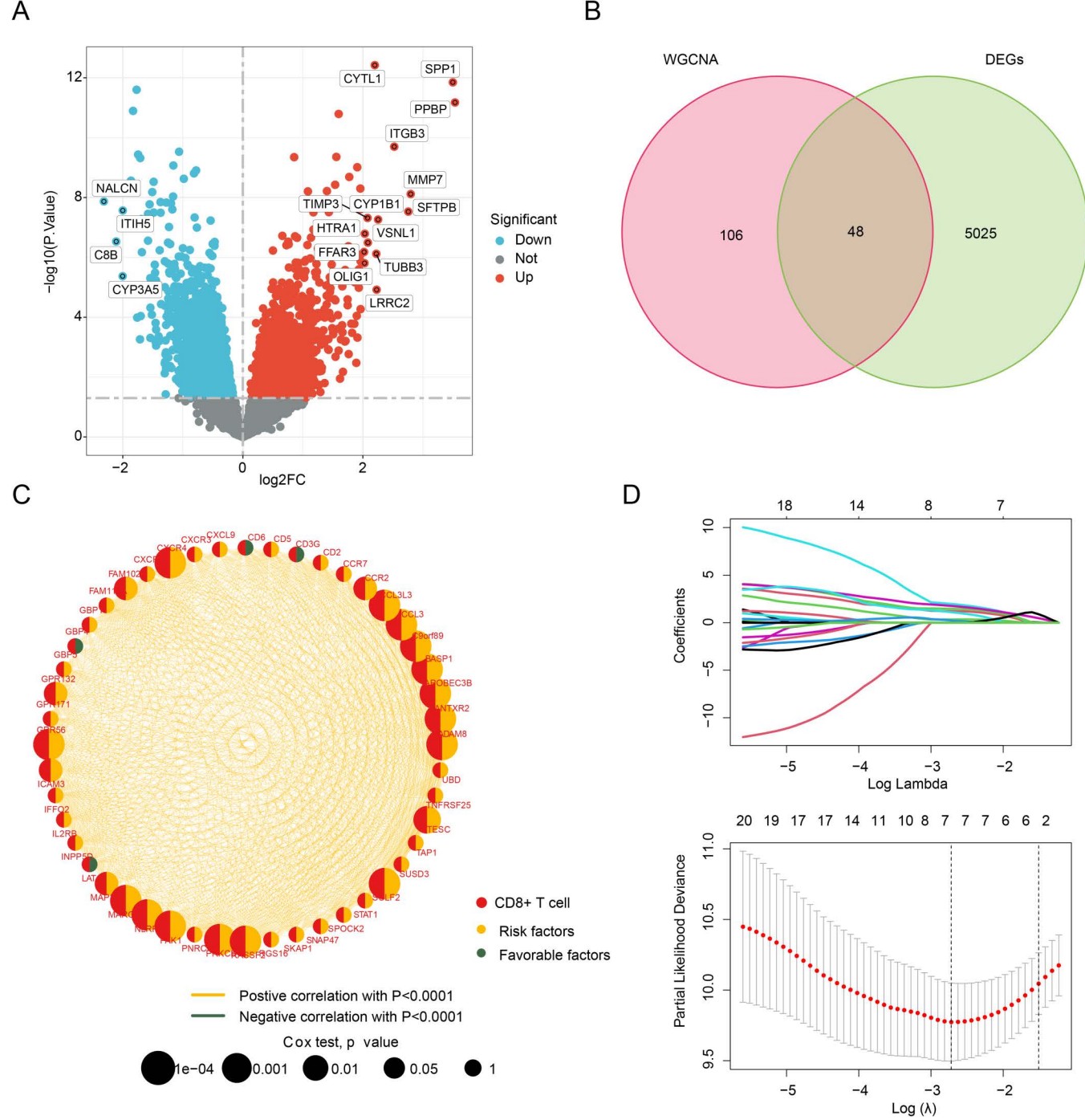

**Fig 2. Identification of CRG related to prognosis. (A) Selection of differentially expressed genes between the normal and IPF groups. (B)** Venn diagram showing the selection of differentially expressed CRG using a WGCNA analysis. **(C)** Network diagram showing the correlation between differentially expressed CRG and prognosis. **(D)** LASSO analysis for feature selection of prognostic variables.

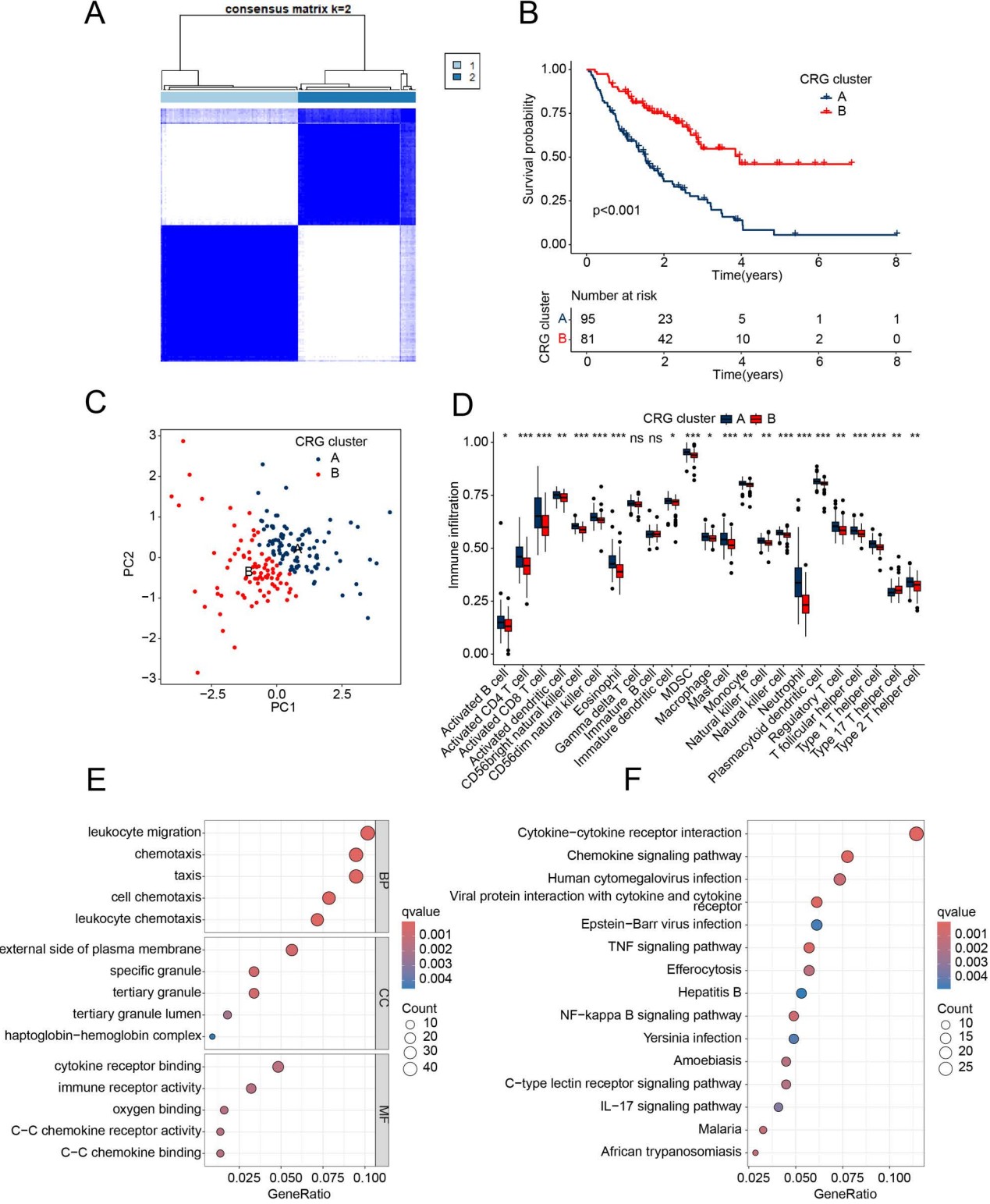

**Fig 3. Identification and functional enrichment analysis of CRG subtypes in IPF. (A)** Analysis of subtypes based on CRG. **(B)** Clinical prognostic analysis of CRG subtypes. **(C)** Unsupervised PCA analysis. **(D)** Immune infiltration features of CRG subtypes. **(E, F)** GO and KEGG analysis of differentially expressed genes in CRG molecular subtypes.

CXCR4 * 2.68 + GPR56 * 2.95 + PAK1 * 4.57. By stratifying IPF samples into risk subgroups based on the optimal cutoff of clinical prognostic outcomes, we discovered that high CRG risk subgroups were more likely to suffer from mortality risk (Fig 4A). Additionally, clinical prognostic outcomes analysis revealed that the low CRG risk subgroup had significantly better clinical prognoses than the high CRG risk subgroup (Fig 4B, $p < 0.001$). These findings suggest that the risk model, developed using three independent prognostic factors, can predict the clinical survival prognosis of IPF risk subgroups. The time-related ROC curve results for 1-, 3-, and 5-year predictions showed AUC of 0.764, 0.713, and 0.845, respectively (Fig 4C). Unsupervised PCA results indicated that the prognostic features based on CRG could clearly distinguish the CRG risk subgroups (Fig 4D). Moreover, we also observed that the expression of CXCR4, GPR56, and PAK1 was significantly upregulated in the high CRG risk subgroup when compared with the low CRG risk subgroup (Fig 4E). In the CRG clustering subgroup, we found that the CRG score was significantly higher in subgroup A, which had a poorer clinical prognosis, than in subgroup B (Fig 4F). In addition, by integrating CRG molecular subtypes, CRG score index, and clinical survival status, we employed a Sankey diagram to comprehensively demonstrate the stratification capability of our model. The results indicated that samples classified as CRG subtype B were more likely to fall into the low CRG score group, while those of CRG subtype A were predominantly associated with the high CRG score group and linked to poor prognosis (Fig 4G). Based on these findings, we highlight the potential association between CRG molecular subtypes and risk scores, further confirming the ability of the CRG scoring model to distinguish survival risks among patients with different immune subtypes. These insights also suggest the potential utility of the CRG score in assisting prognostic evaluation and guiding personalized therapeutic strategies.

## Validation the accuracy of CRG risk score in predicting clinical prognosis of IPF

Based on the "caret" R script, IPF samples from GSE70866 were randomly assigned to two independent sets – a training set and a validation set. A CRG risk score was developed to assess the independence and accuracy of the CRG risk model in predicting the clinical prognosis of IPF. Using the best cutoff value for clinical prognosis, IPF samples in the two independent sets were classified into high and low CRG risk subgroups. The survival curve results indicated that in the two independent sets, IPF samples with high CRG risk scores had a worse clinical prognosis survival outcome than those with low CRG scores (Fig 5A, 5B). Additionally, the time-related ROC curve results showed that the AUC for 1-, 3-, and 5-year in the training set were 0.757, 0.729, and 0.834, respectively, and in the validation set were 0.733, 0.629, and 0.909, respectively (Fig 5C, 5D). Furthermore, unsupervised PCA analysis in the two independent sets revealed two significantly different distribution patterns in the CRG risk subgroups (Fig 5E, 5F). Based on these findings, we conclude that the risk model constructed based on CRG prognostic features accurately evaluates the clinical survival outcome of IPF.

## Independent prognostic analysis of CRG score

Given that the CRG score can accurately assess the clinical survival outcomes of IPF, we conducted further evaluations to determine its independent prognostic value. We subsequently used univariate and multivariate Cox analyses to comprehensively evaluate the HR values of the CRG score and other clinical pathological features, as demonstrated in Fig 6A–6B. The results of the univariate Cox analysis suggested that gap (HR = 1.395 (1.237–1.574), $p < 0.001$) and CRG score (HR = 1.062 (1.034–1.091), $p < 0.001$) were closely associated with adverse prognosis in IPF. The multivariate Cox analysis results indicated that the CRG score was an independent prognostic factor for IPF. The ROC curve results revealed that the AUC of the CRG score was 0.764, which was significantly higher than other clinical pathological features of IPF, demonstrating a high model diagnostic ability (Fig 6C). Furthermore, by integrating clinical feature parameters and the CRG score, we developed a nomogram model to evaluate the 1-, 3-, and 5-year prognostic probabilities of IPF samples (Fig 6D). In summary, our findings suggest that the risk model developed based on CRG prognostic features is an independent prognostic factor for IPF that distinguishes it from clinical pathological features and can be used to accurately predict the survival outcomes of IPF.

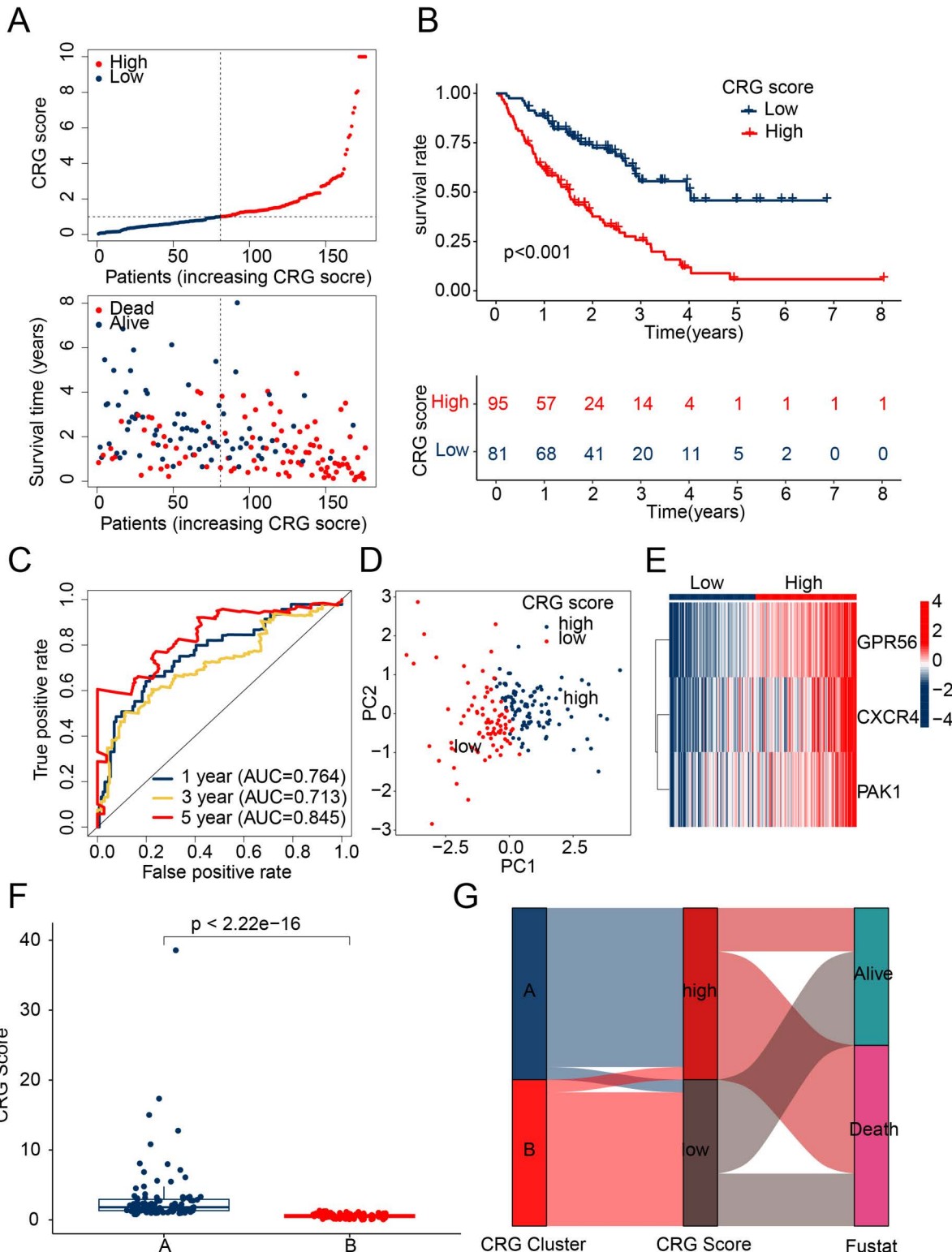

**Fig 4. CRG risk score evaluation and prognostic model construction. (A)** Stratification of CRG into risk subgroups. **(B)** Clinical prognosis outcome analysis of CRG risk subgroups. **(C)** Time-related ROC curve analysis. **(D)** PCA analysis based on CRG prognostic features. **(E)** Expression profile analysis of three independent prognostic factors in CRG score subgroups. **(F)** Analysis of differences in CRG risk scores in CRG clustering subgroups. **(G)** Sankey diagram showing the association between CRG clustering subgroups, risk subgroups, and clinical prognosis.

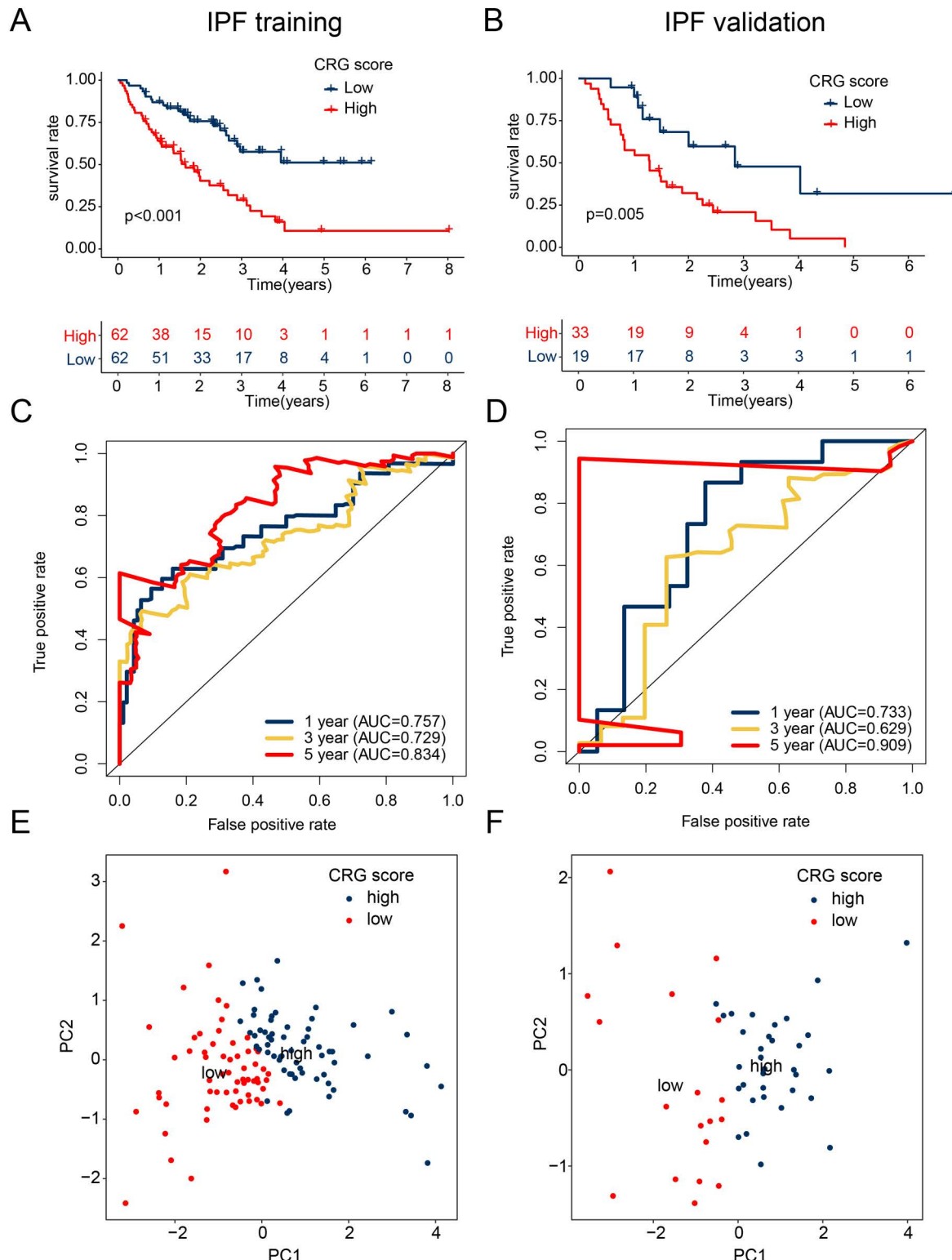

**Fig 5. Risk model construction for CRG in independent cohorts. (A, B)** Clinical prognostic outcome analysis of the training and validation sets of IPF. **(C, D)** Analysis of time-dependent ROC curves in two independent cohorts. **(E, F)** PCA analysis of the training and validation sets.

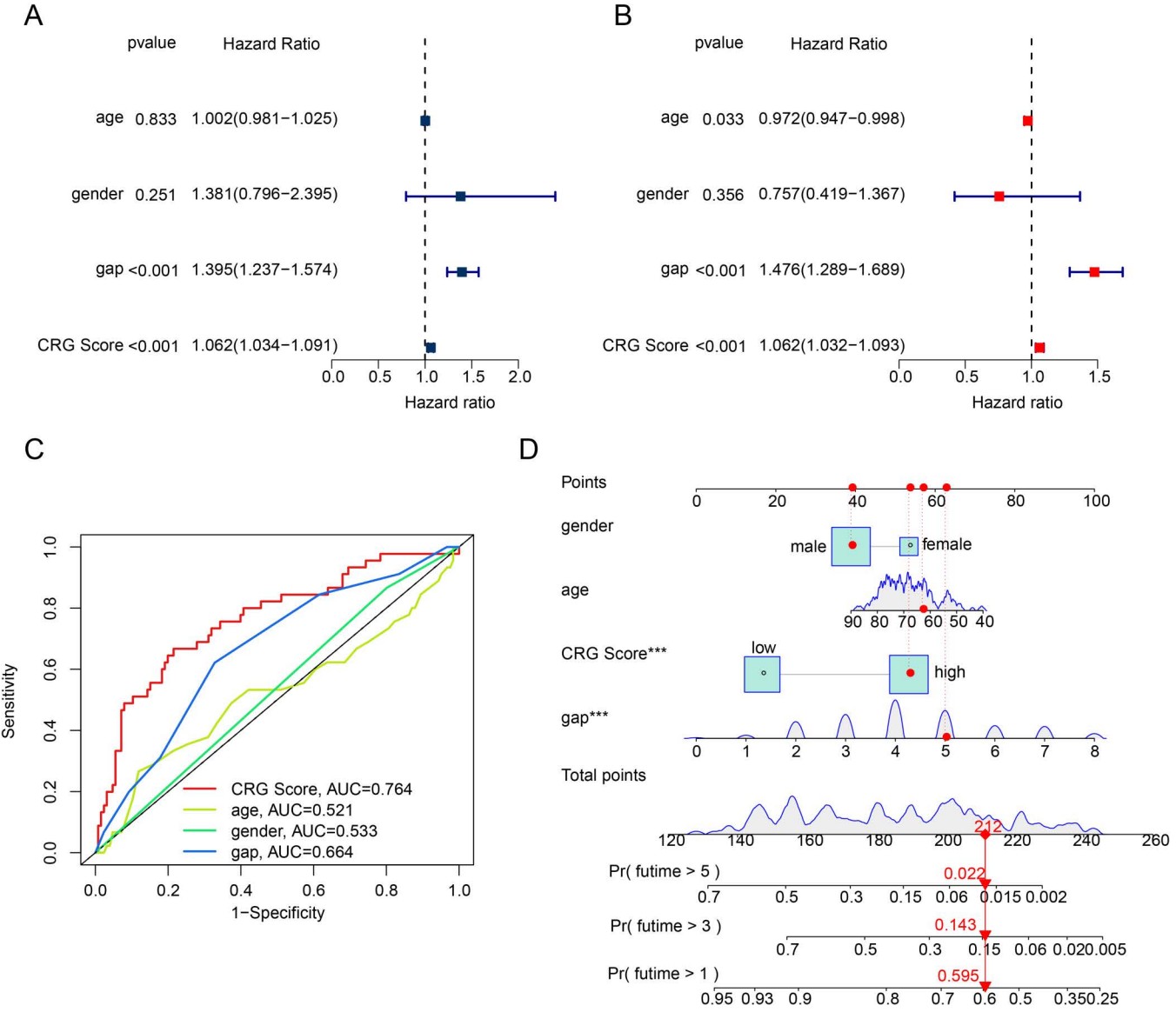

**Fig 6. Independent prognostic analysis of CRG score and various clinical pathological features. (A, B)** HR and *p* values of CRG score and different clinical pathological features were evaluated based on univariate and multivariate Cox analyses. **(C)** Diagnostic effectiveness evaluation of CRG score and clinical pathological features. **(D)** Development of a nomogram model based on CRG score and clinical pathological parameters.

## Immune infiltration characteristics analysis of CRG score subgroups

Based on the GSVA algorithm, we assessed the KEGG signaling pathways of each IPF sample in the CRG risk subgroups to investigate potential mechanisms between the two subgroups. Our GSVA findings revealed that the beta-alanine metabolism signaling pathway was significantly upregulated in the low CRG risk subgroup. Notably, we also observed significant upregulation of immune-related signaling pathways in the high CRG risk subgroup, including the Chemokine signaling pathway, T cell receptor signaling pathway, B cell receptor signaling pathway, and NOD-like receptor signaling pathway. This suggests that immune-related molecular functions may play a key role in regulating the CRG risk subgroup (Fig 7A). Immune infiltration results further supported this finding, with a significantly higher proportion of immune cells

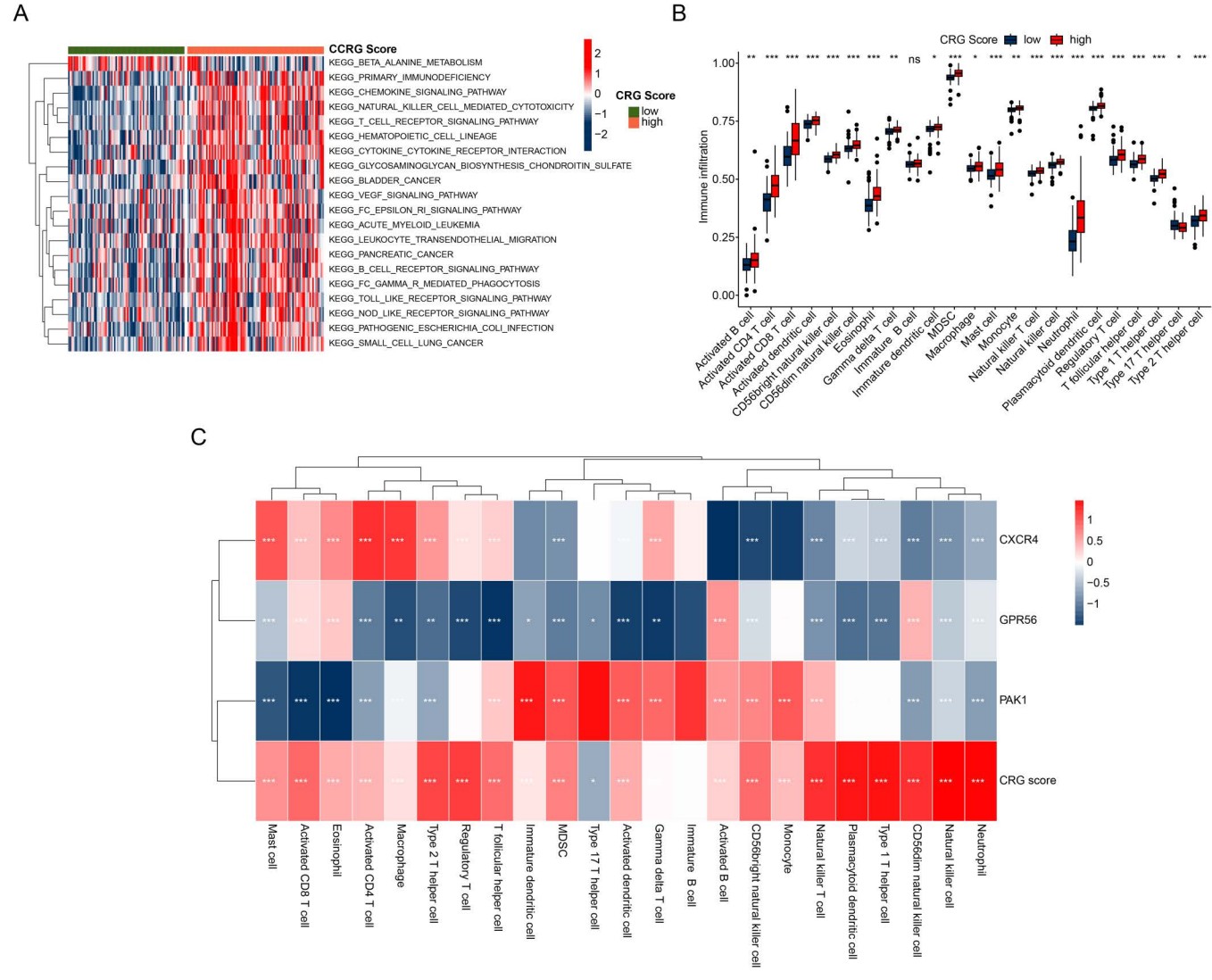

**Fig 7. The immune infiltration characteristics between the high and the low CRG risk subgroups. (A)** shows the KEGG terms identified in the CRG score subgroups using the GSVA algorithm. **(B)** Immune cell infiltration landscape in the CRGs score subgroups. **(C)** The potential relationship between CRG score and immune cells.

in the high CRG risk subgroup compared to the low CRG risk subgroup, indicating a higher immune state in the former (Fig 7B). Correlation analysis revealed potential associations between three independent prognostic factors and 23 immune cells in the CRG score, with the CRG score showing significant positive correlation with most immune cells. We also observed significant correlations between CXCR4, GPR56, PAK1, and most immune cells (Fig 7C). In summary, our results suggest that the immune status of the high CRG risk subgroup with poor prognosis is higher than that of the low CRG risk subgroup, with the clustering subgroup results showing consistency.

### *In vitro* validation of the expression of CRG prognostic signature in IPF cell models

In subsequent experiments, we further validated the mRNA and protein expression patterns of the CRG prognostic signature in an *in vitro* cellular model. The MRC-5 cell line was used as the control, while the IPF cell model was established

by stimulating MRC-5 cells with TGF-β1 (3 μg/L) for 48 hours. qRT-PCR analysis revealed that, compared to the control group (MRC-5), the mRNA expression levels of CXCR4, GPR56, and PAK1 were significantly upregulated in the model group (MRC-5+TGF-β1), consistent with the results of our previous bioinformatics analysis (Fig 8A–8C).

Furthermore, Western blot analysis was conducted to assess the protein expression levels of the CRG prognostic signature in both control and model groups. Quantitative analysis of the Western blot data indicated that the protein levels of CXCR4, GPR56, and PAK1 were markedly elevated in the model group (Fig 8D, 8E). These findings provide experimental support for the reliability of the CRG prognostic signature and its potential involvement in the pathogenesis of IPF.

## Discussion

In this study, we screened IPF prognosis-associated CRGs, and three independent prognostic factors (*CXCR4*, *GPR56*, and *PAK1*) were used to establish the prognostic signature of IPF. In recent years, many new therapeutic targets for IPF have been developed. However, most clinical trials ultimately failed in patients with IPF, suggesting the need for further exploration of therapeutic targets [19]. At the same time, the precise stratification of the risk of IPF is one of the difficulties

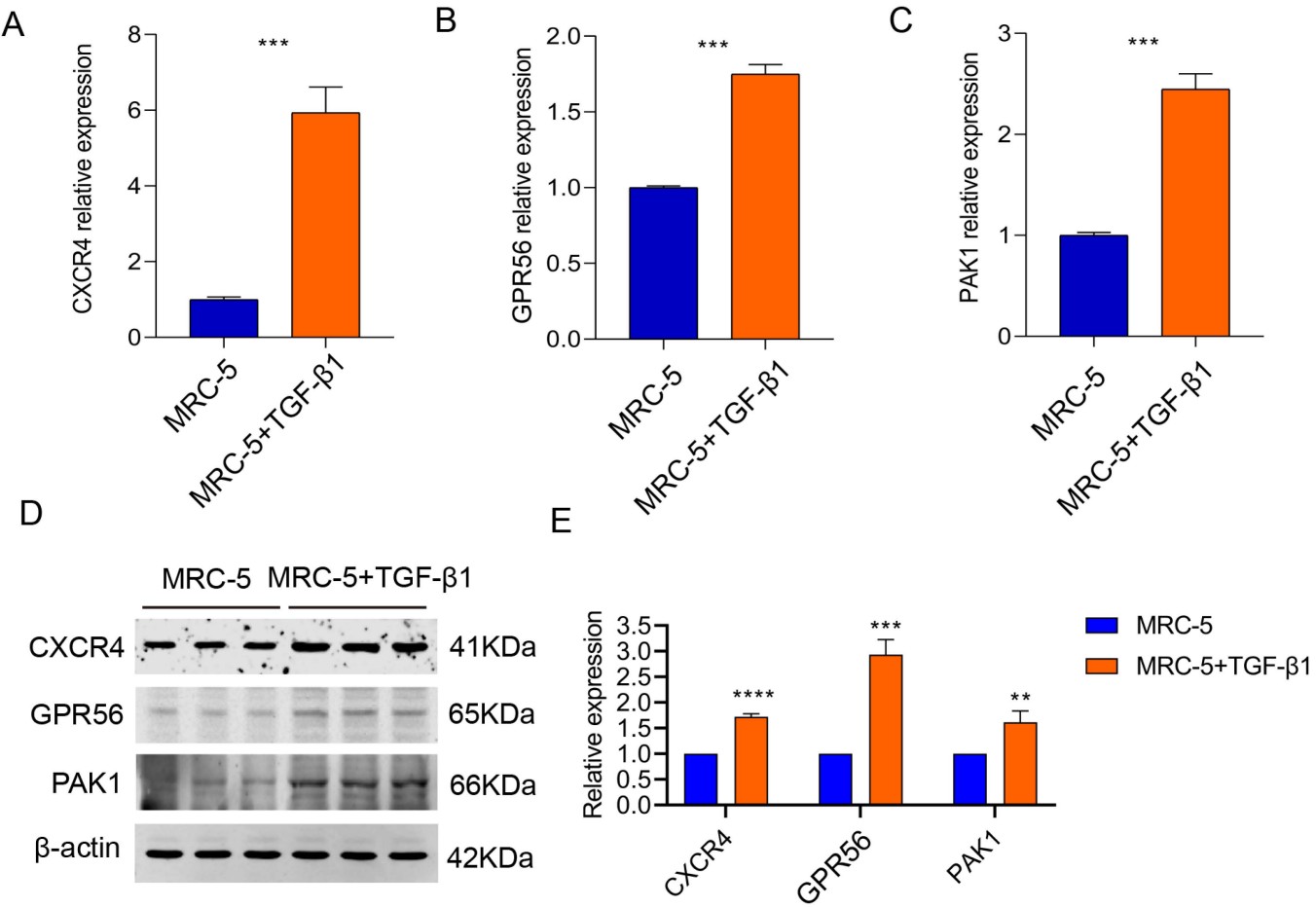

**Fig 8. qRT-PCR and western blot analysis of prognostic CRG variables in cell lines.** mRNA expression of **(A)** CXCR4, **(B)** GPR56, and **(C)** PAK1 in MRC-5 and MRC-5+TGF-β1 cell lines (n = 3). Western blot analysis of the expressions of CXCR4, GPR56, and PAK1 in MRC-5 cell and MRC-5+TGF-β1 cell lines (n = 3). Data are presented as the mean ± SD. $*p < 0.05$; $**p < 0.01$; $***p < 0.001$.

of treatment [20]. The prognostic features of IPF established in this study reflect the role of immune cells in IPF, and three potential therapeutic targets have been screened out and are helpful for the prognosis assessment of patients with IPF.

Our molecular typing results show that there are significant differences in prognosis and immune microenvironment between the two. Pathway enrichment analysis suggested that the chemotaxis related pathway might be part of the reason for the differences between the two. It is important to note that the gene expression data in GSE70866 were derived from BAL cells rather than lung tissue biopsies. The expression of immune-related genes observed in our study may primarily reflect changes in the immune microenvironment of the lower respiratory tract rather than direct alterations in lung parenchymal cells or fibrotic tissue. Chemotaxis induced by BAL play an important role in the development of IPF [21]. There is evidence suggesting that the elevated levels of interleukin-8 (IL-8) and monocyte chemoattractant protein 1 (MCP-1) may be characteristic of IPF [22]. As a classic monocyte/macrophage chemokine, the activation of MCP-1 can aggravates IPF [23]. IL-8 is a potent chemotactic agent for neutrophils and can also indicate the disease activity of IPF [24]. These findings further support the critical role of chemotaxis in the pathogenesis of IPF and highlight IL-8 and MCP-1 as potential therapeutic targets for modulating the immune microenvironment in the lower respiratory tract.

During the process of molecular subtyping, we found that the expression of regulatory T cells (Tregs) and natural killer cells (NK cells) was significantly correlated with prognosis. Previous single-cell RNA sequencing studies have revealed that increased numbers of classical monocytes and Tregs in IPF are closely associated with disease prognosis [25,26]. Further research demonstrated that CD14+CD163-HLA-DRlow monocytes are most strongly correlated with disease progression and also impact the prognosis of IPF patients [1,27]. These monocytes appear to promote the expansion of Tregs while suppressing the function of NK cells and certain T-cell subsets [28]. Peripheral exhaustion of NK cells and imbalance of the Treg/Th17 axis in patients with IPF are its important characteristics [29]. This may explain why the expression levels of Tregs and NK cells are associated with IPF progression. Given the significant differences in the expression levels of various immune cells within the immune microenvironment among different subtypes-which are linked to distinct patient outcomes-the complex immune microenvironment in IPF patients may be a critical factor influencing disease progression.

Among the three independent prognostic factors screened, CXCR4 is a transmembrane protein receptor involved in stem cell migration, which is mainly expressed in epithelial cells and bone marrow cells [30]. The role of CXCR4 and its ligand CXCL12 in organ fibrosis has been demonstrated [31]. Increased CXCR4 cells can be observed in lung tissue of patients with IPF [32]. In the area of fibrotic remodeling in IPF patients, immunohistochemical analysis showed higher CXCR4 staining rates in epithelial cells and macrophages, suggesting that the CXCR4/ CXCL12 axis may be an important mechanism for fibrosis development [33]. Because CXCR4 inhibition is a well-tolerated approach in clinical use [34], targeting epithelial CXCR4 may be a suitable treatment strategy for IPF. The mTOR inhibitor Sirolimus inhibits CXCR4 expression in fibroblasts. A short-term randomized double-blind trial showed that short-term treatment with Sirolimus reduced circulating fibroblast concentration in IPF subjects with a good safety profile [35]. Therefore, CXCR4 is expected to be a key target to overcome IPF.

GPR56 encodes an orphan G-protein-coupled receptor (GPCR) that is widely expressed in the nervous system and is essential for the normal development of cerebral cortex and cerebellar morphology [36,37]. At present, the research on GPR56 is in its infancy, and there have been no reports on IPF. However, various GPCRS have been reported to be associated with the pathogenesis of pulmonary fibrosis through the promotion of profibrotic fibroblast activation, and several clinical trials have been conducted as potential therapeutic targets [38]. As a member of the GPCR family, GPR56 has a long N-terminal extracellular region [39] that interacts with multiple ligands. The screening of GPR56 in this study and the high expression of GPR56 in IPF model indeed provide a new target for further research in the future.

PAK1 kinase is an effector of Cdc42 and Rac1 GTP enzymes that regulates cell projection and movement by controlling actin and adhesion dynamics [40]. There is no direct evidence linking PAK1 to IPF. One possibility is through the AKT signaling pathway. The effect of Akt on extracellular signal-regulated kinases depends on the status of PAK1 [41,42].

Inhibition of PAK1 leads to continued activation of AKT [43]. The interaction of AKT with transforming growth factor-β (TGF-β) promotes the formation of pulmonary fibrosis [44]. The AKT pathway is involved in pulmonary fibrosis by regulating its downstream, such as mTOR, HIF-1α, and the FOX family [45,46]. Targeting AKT pathway is expected to be a new strategy for IPF treatment [47]. Whether the abnormal expression of PAK1 in IPF patients and models is related to the AKT pathway needs further study.

Importantly, in this study, the prognostic genes were identified from transcriptomic data derived from BAL cells, which predominantly reflect the immune landscape of the fibrotic lung [48]. However, *in vitro* validation was conducted in MRC-5 fibroblasts, a mesenchymal cell line commonly used to model fibroblast activation. Although this may appear as a limitation, it also reflects the complex intercellular communication underlying IPF pathogenesis, in which immune cells and fibroblasts are engaged in dynamic cross-talk [49–51]. Moreover, immune cells and fibroblasts may also have a common activation pathway in IPF [52,53]. The further establishment of an *in vitro* co-culture model of immune cells and fibroblasts in the future will help us deepen our understanding of the connection between the two.

Immuno-infiltration test results showed that patients with higher CRG scores had significantly higher levels of most immune cells compared to patients with lower scores. In addition, when comparing the results of pathway enrichment analysis in patients with high and low scores, NK cell mediated cytotoxicity, T cell receptor signaling pathway, B cell receptor signaling pathway and other pathways are highly enriched in patients with high scores. Combined with the poor prognosis of patients with higher CRG scores, this suggests that immune cells play a role in IPF. We also found that beta-alanine metabolism had lower enrichment in patients with higher CRG scores. beta-alanine and its metabolite carnosine play an important role in the body as a buffer and antioxidant [54]. Restoration of antioxidant system plays an important role in IPF treatment [55]. The potential protective effect of beta-alanine on IPF deserves further study.

This study has several limitations. One of the limitations of this study is the lack of detailed clinical parameters in the GSE70866 dataset. Although our prognostic model based on CRGs showed significant stratification of overall survival, we were unable to evaluate its correlation with key indicators of IPF severity and progression, such as forced vital capacity (FVC), diffusion capacity for carbon monoxide (DLCO), 6-minute walk distance (6MWD), acute exacerbations, or relevant comorbidities, due to the unavailability of such data. These parameters are critical for assessing functional impairment and disease dynamics in IPF and would have provided valuable insight into the clinical utility and independence of the CRGs score. Therefore, the absence of these data limits our ability to perform a more comprehensive clinical validation of the proposed molecular risk model. Future studies using datasets with complete lung function and follow-up information will be essential to validate and refine the prognostic relevance of CRGs in clinical settings. Second, although our findings highlight the prognostic value of CRGs, the study lacks in-depth mechanistic exploration of how these genes regulate immune responses or contribute to IPF progression. Third, functional validation was limited to *in vitro* assays using cell lines, which do not fully capture the complexity of the IPF microenvironment, particularly the interactions between immune cells and fibrotic tissue. Future research should focus on several key directions to address these limitations: *In vivo* experiments using established models of pulmonary fibrosis are needed to validate the functional roles of these genes under physiological conditions; Incorporation of large, well-annotated clinical cohorts with comprehensive lung function data and longitudinal follow-up will be crucial to further evaluate and refine the prognostic utility of the CRG score.

## Supporting information

**Supplementary Table 1.  Clinical baseline characterization of GSE70866 dataset.**
(XLSX)

**Supplementary Table 2.  Identification of CRG based on WGCNA algorithm.**
(XLSX)

**S1 File.**
(PDF)

## Author contributions

**Investigation:** Lin Feng, Jiabo Yuan.

**Software:** Lin Feng.

**Supervision:** Xiaobing Dou, Yanlin Liu.

**Validation:** Yanlin Liu.

**Writing – original draft:** Xiaobing Dou, Yanlin Liu.

**Writing – review & editing:** Xiaobing Dou, Yanlin Liu.

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
