## [Decision Letter · Decision Letter 0]

PONE-D-25-16281Comprehensive analysis of molecular characteristic and clinical prognosis of CD8+ T cell-associated genes in Idiopathic pulmonary fibrosisPLOS ONE

Dear Dr. Liu,

Thank you for submitting your manuscript to PLOS ONE. After careful consideration, we feel that it has merit but does not fully meet PLOS ONE’s publication criteria as it currently stands. Therefore, we invite you to submit a revised version of the manuscript that addresses the points raised during the review process.

We look forward to receiving your revised manuscript.

Kind regards,

Simone Agostini, Ph.D.

Academic Editor

PLOS ONE

4. Please note that PLOS ONE has specific guidelines on code sharing for submissions in which author-generated code underpins the findings in the manuscript. In these cases, we expect all author-generated code to be made available without restrictions upon publication of the work. Please review our guidelines at https://journals.plos.org/plosone/s/materials-and-software-sharing#loc-sharing-code and ensure that your code is shared in a way that follows best practice and facilitates reproducibility and reuse.

5. We note that your Data Availability Statement is currently as follows: [All relevant data are within the manuscript and its Supporting Information files.]

Reviewers' comments:

Reviewer's Responses to Questions

**Comments to the Author**

1. Is the manuscript technically sound, and do the data support the conclusions?

Reviewer #1: Yes

Reviewer #2: Yes

2. Has the statistical analysis been performed appropriately and rigorously? 

Reviewer #1: Yes

Reviewer #2: Yes

3. Have the authors made all data underlying the findings in their manuscript fully available?

Reviewer #1: Yes

Reviewer #2: Yes

4. Is the manuscript presented in an intelligible fashion and written in standard English?

Reviewer #1: Yes

Reviewer #2: Yes

5. Review Comments to the Author

Reviewer #1: This manuscript addresses a clinically important topic and combines different bioinformatics approaches with experimental validation, which strengthens its translational relevance. The use of multiple statistical and machine learning techniques, such as WGCNA, LASSO, Cox regression, consensus clustering) is appropriate and well-aligned with the study goals. The paper is well written, the data is well presented and analyzed. However, the following general points should be addressed.

• The rationale for selecting CD8+ TCRGs and the source of the gene list is not clearly described.

• The manuscript would benefit from a clearer explanation of how the prognostic model was validated, particularly regarding the use of training and validation sets.

• The in vitro validation is a strong point, but more detail is needed on experimental replicates, controls, and statistical analysis.

• The manuscript should better clarify whether the identified markers are novel in the context of IPF or confirmatory of previous findings.

See the uploaded document for the specific major and minor comments.

Reviewer #2: Thank you for the opportunity to review this well conducted and very interesting study that further supports the Role of immunity in IPF. This study was an unmet need. While new data were available for monocytes and Tregs in IPF, there was a need for such studies for CD8 T cells. The analysis is elegant and the figures are nice ( volcano plots, Survival curves, heat maps etc all of them are elegant). Some comments that could even improve more this manuscript are provided. They mainly reflect the effort to identify therapeutic targets and to further improve the discussion based on other recent works for immunity in IPF. Specific comments follow:

1) Authors should do connectivity map analysis using as input the CD8 related genes that they found to find drugs that increase or decrease these genes. This can help the field move on. For example they can use the Clue platform for that which accepts as input up to 250 genes ( in case the genes are more, those that have the higher log2fold change are selected).

2) Authors should substantially improve their discussion. First of all, it would be better to start it with the main findings of this work and not with introductory sentences. Secondly authors need to discuss more immune abberrations in IPF and how their work is relevant compared to other recent works. Single cell RNA sequencing studies in IPF found an outcome-predictive increase in classical monocytes and T regulatory cells. Further investigation showed that CD14+CD163–HLA-DRlow monocytes were the monocytes that were most consistently associated with progression. These monocytes seem to be associated with expansion of Tregs, suppression of natural killer cells and suppression of some T cell sub populations. This could be the reason that expansion of Tregs, and suppression of natural killer cells have been associated with IPF progression.

Further research aiming to address whether CD8 T cells are part of the same cascade and for their exact interaction with other immune subpopulations is greatly anticipated.

The studies that describe the above are provided here and should be in the reference list along with a more detailed presentation of immunity in IPF in the discussion.

https://pubmed.ncbi.nlm.nih.gov/38717443/

https://www.medrxiv.org/content/10.1101/2024.08.07.24311386v2

https://pmc.ncbi.nlm.nih.gov/articles/PMC8491266/

https://pmc.ncbi.nlm.nih.gov/articles/PMC12001815/

https://pmc.ncbi.nlm.nih.gov/articles/PMC3570653/

https://www.thelancet.com/journals/ebiom/article/PIIS2352-3964(23)00332-8/fulltext

https://pubmed.ncbi.nlm.nih.gov/39510135/

3) Authors should change a title of the results by writing “Validation of the accuracy” instead of “Validate the accuracy of” so that they are consistent with the other titles of results

6. PLOS authors have the option to publish the peer review history of their article (what does this mean? ). If published, this will include your full peer review and any attached files.

**Do you want your identity to be public for this peer review?** For information about this choice, including consent withdrawal, please see our Privacy Policy .

Reviewer #1: No

Reviewer #2: No

---

## [Author Response · Author response to Decision Letter 1]

18 Jun 2025

We sincerely thank the editor and reviewers for their constructive comments and suggestions. We have carefully revised our manuscript accordingly. Below are our point-by-point responses to each comment.

PONE-D-25-16281

Title of the manuscript: Comprehensive analysis of molecular characteristic and clinical prognosis of CD8+ T cell-associated genes in Idiopathic Pulmonary Fibrosis

1: Summary of the manuscript:

The present study explores the role of CD8+ T cells in idiopathic pulmonary fibrosis (IPF) by identifying CD8+ T cell-associated genes associated with the prognosis (mainly survival) of the disease. Using transcriptomic data from the GSE70866 dataset, the authors apply WGCNA, differential expression analysis, and Cox regression to identify key prognostic markers – focusing on CXCR4, GRP56, and PAK1. These markers are used to construct a prognostic model and molecular subtypes of IPF. The model is validated using survival analysis and ROC curves. In addition, the results in-silico were validated in vitro using a TGF-β1-induced IPF cellular model, which confirmed the altered expression of the identified genes at both mRNA and protein levels. All in all, the study aims to provide new insights into IPF pathogenesis and potential prognostic biomarkers.

2: General comments

This manuscript addresses a clinically important topic and combines different bioinformatics approaches with experimental validation, which strengthens its translational relevance. The use of multiple statistical and machine learning techniques, such as WGCNA, LASSO, Cox regression, consensus clustering) is appropriate and well-aligned with the study goals. The paper is well written, the data is well presented and analyzed. However, the following general points should be addressed.

1: The rationale for selecting CD8+ TCRGs and the source of the gene list is not clearly described.

Reply

Thank you very much for this insightful and critical comment. Previous studies have demonstrated that alterations in the immune microenvironment play a crucial role in the onset and progression of IPF. Therefore, a central aim of our study was to systematically investigate the potential associations between the immune microenvironment composition and IPF.

To begin with, we employed the ssGSEA algorithm to quantitatively analyze the infiltration levels of 23 immune cell types, aiming to identify those most closely associated with IPF. Subsequently, we utilized the WGCNA algorithm to assess co-expression relationships between these immune cell types and the IPF expression profile. Our results revealed that CD8+ T cells exhibited the strongest correlation with IPF, suggesting that they may represent a key immune component involved in the progression of the disease.

Building on this finding, we extracted a core gene signature from the WGCNA module most strongly associated with CD8+ T cells and conducted further in-depth analyses of these genes. In the subsequent steps, we evaluated the prognostic relevance of these genes in IPF patients and developed a CD8+ T cell-related gene (CRG) signature-based prognostic model, referred to as the CRG score, to stratify patient risk.

Once again, we sincerely appreciate the reviewer for pointing out the lack of clarity in this section. In the revised manuscript, we have added and clarified detailed information regarding the genes included in this module to enhance the logical coherence and readability of the study.

Supplementary table 2, Line 104

2: The manuscript would benefit from a clearer explanation of how the prognostic model was validated, particularly regarding the use of training and validation sets.

Reply

We sincerely thank the reviewer for the valuable comment. We fully agree with your suggestion and have revised the manuscript to provide a more detailed and clearer description of the construction and validation process of the prognostic model.

In this study, we used the “caret” R package to randomly divide all IPF patient samples into a training set and a validation set at a 7:3 ratio. The training set was utilized to perform univariate Cox regression, LASSO regression, and multivariate Cox regression analyses to construct the prognostic risk score model. The validation set was then used to evaluate the robustness and predictive performance of the constructed model.

In both independent cohorts, we generated Kaplan-Meier survival curves and time-dependent ROC curves, and compared their AUC values to assess the model’s consistency and accuracy across datasets.

We believe that these additions help to improve the overall clarity and persuasiveness of the manuscript. Once again, we are grateful for your insightful suggestions.

Line 107-125

3: The in vitro validation is a strong point, but more detail is needed on experimental replicates, controls, and statistical analysis.

Reply

We sincerely thank the reviewer for the positive evaluation of our in vitro experiments and for the thoughtful and constructive suggestions for improvement. In the revised manuscript, we have provided more detailed descriptions in the Methods section, as outlined below:

1. Experimental reproducibility: All in vitro experiments were independently repeated at least three times to ensure the reliability and reproducibility of the results.

2. Control settings: In our cell experiments, we established clear control (MRC-5) and model (MRC-5 + TGF-β1) groups to evaluate the changes in mRNA and protein expression levels of the target gene under different conditions.

3. Statistical analysis: All statistical analyses were performed using GraphPad Prism 8.0. Two-tailed Student’s t-tests were used for comparisons between two groups, and significance levels (P < 0.05, P < 0.01, P < 0.001) were clearly indicated in the figure legends. Data were presented as mean ± standard deviation (SD).

We believe that these additions help to improve the rigor and scientific credibility of the in vitro experimental section. Once again, we are grateful to the reviewer for their careful evaluation and valuable feedback.

Line 172-183, line 329-342, Figure 8

4: The manuscript should better clarify whether the identified markers are novel in the context of IPF or confirmatory of previous findings.

Reply

We sincerely thank the reviewer for the valuable suggestions provided for our study. In response to your comments, we have expanded the Discussion section to include a more comprehensive interpretation of the key biomarkers identified in the context of IPF, and further elaborated on the novelty of our findings.

Specifically, we conducted a systematic literature review and comparative analysis of the previously reported roles of the selected genes in IPF-related studies. Our results indicate that PAK1 and GPR56 have not been previously reported to be associated with IPF, making our study the first to propose their potential involvement in the pathogenesis of the disease. This highlights the novelty and exploratory value of our findings. In contrast, CXCR4 has been documented in prior studies, which confirmed its upregulation in lung tissues of IPF patients and suggested its potential role in disease progression and in predicting responses to pirfenidone treatment (Relevant references: PMID: 29507348, 32843095, 32822674, 29453386). Our results further validate and support the functional relevance of CXCR4 in IPF.

We believe that these additions enhance the logical flow and comprehensiveness of the Discussion section, while also better emphasizing the academic value and significance of our study. Once again, we thank the reviewer for their attention to and guidance on our work.

Line 381-3894, Line 391-393

3�Specific comments

Major comments:

1: Clarification of type of samples used: The manuscript does not clearly specify that the transcriptomic data from the GSE70866 dataset comes from bronchoalveolar lavage (BAL) cells. This is a critical detail that should be explicitly stated in the material and methods (IPF data set collection, line 77) and discussed in the context of the findings, as it influences the interpretation of immune-related gene expression and the relevance to lung tissue pathology in IPF.

Reply

We sincerely appreciate the reviewer’s professional and important suggestion. In response to this concern, we have explicitly clarified the source of the GSE70866 dataset in the Materials and Methods section, stating that it comprises transcriptomic data derived from bronchoalveolar lavage (BAL) cells of patients with IPF.

Additionally, we have expanded the Discussion section to further explain the rationale for using BAL cells as samples for immune profiling. We have briefly discussed both their representativeness and limitations in elucidating the immunopathological mechanisms of IPF, aiming to help readers better understand the study context and interpret the results more accurately.

We believe that these revisions enhance the scientific rigor and readability of the manuscript. Once again, we thank the reviewer for their thorough evaluation and constructive feedback.

Line 351-366

2: Sample matching (IPF and controls): The manuscript describes the use of SVA and limma for batch correction and normalization, which is appropriate. However, it is unclear whether any matching or adjustment was performed to account for potential confounding variables, such as age or sex, between IPF and control groups. The authors should clarify whether such matching was considered or whether the groups were balanced.

Reply:

We sincerely thank the reviewer for raising this critical and insightful question. We fully agree that controlling for potential confounding factors such as age and sex is essential to ensure the reliability and scientific validity of group comparison analyses.

In this study, prior to data integration and downstream analyses, we carefully reviewed the clinical information of the included datasets, with particular attention to the distribution of age and sex between the IPF and normal control groups. It is important to note that although the two platform files were generated from different sources, they both belong to the same GEO dataset and include complete clinical feature information for all samples. Preliminary assessments indicated that the distribution of age and sex was relatively balanced between groups.

To further minimize potential bias from platform batch effects and confounding variables, we applied standardization and batch effect correction using the SVA package. These methodological details have been added to the revised manuscript. Additionally, we have included baseline clinical characteristics of the samples in the Materials and Methods section and provided them as Supplementary Table 1, to improve the transparency and reproducibility of our data processing.

We believe these revisions strengthen the rigor and persuasiveness of our study. Once again, we sincerely thank the reviewer for the thoughtful review and valuable suggestions.

Supplementary table 1, line 83-line 93

3� Interpretation of WGCNA results: The authors identify and focus on the cyan module as the most correlated with CD8+ T cells (r = 0.85, p = 2e-56). However, the biological interpretation of this module is not discussed. What types of genes are enriched in this module? Are there known pathways or functions?

Reply:

We sincerely thank the reviewer for the valuable comments. In response to your suggestion, we have conducted a comprehensive functional enrichment analysis of the genes within the cyan module to further explore their potential biological significance.

Specifically, we performed Gene Ontology (GO) and Kyoto Encyclopedia of Genes and Genomes (KEGG) enrichment analyses. The results revealed that the genes in this module are significantly enriched in various immune-related biological processes and signaling pathways, including the T cell receptor signaling pathway, T cell activation, and chemokine signaling pathway. These findings suggest that the cyan module may play a critical role in CD8⁺ T cell-mediated immune responses, thereby further supporting the scientific rationale and validity of using this module as the basis for constructing our prognostic model.

These functional annotations have been added to the revised manuscript in the Results section. Once again, we sincerely thank the reviewer for the professional suggestion, which has substantially enhanced the depth and rigor of our study.

Figure 1E, F, line 196-205

4�Molecular Subtyping: The authors identify two molecular subtypes (A and B) based on CD8+ TCRGs expression, with subtype A associated with worse clinical prognosis survival and higher immune infiltration. While the clustering appears statistically sound, the biological implications of these subtypes are not fully explored.

Reply

We sincerely appreciate the reviewer’s interest in our molecular subtype analysis and the constructive suggestions provided. In response to your comments, we have conducted a more in-depth and systematic comparative analysis of the biological characteristics of subtype A and subtype B during the revision process.

Specifically, we evaluated the differences between the two subtypes in terms of key signaling pathway enrichment and immune cell infiltration levels. The results revealed that subtype A not only exhibited a poorer survival outcome but also showed significant enrichment of its signature genes in several canonical immune-related pathways, such as leukocyte migration, chemotaxis, immune receptor activity, cytokine–cytokine receptor interaction, and chemokine signaling pathway. These findings suggest that subtype A may be in a state of “immune activation accompanied by immune exhaustion,” a feature that has also been observed in certain autoimmune diseases and tumor immune microenvironments.

Furthermore, we employed the single-sample Gene Set Enrichment Analysis (ssGSEA) method to assess the distribution of various immune cell types across the two subtypes. The results demonstrated that subtype A had significantly higher infiltration levels of CD8⁺ T cells, B cells, and CD4⁺ T cells, further supporting the hypothesis of an immune-activated state in subtype A.

These results not only help to explain the potential immunological mechanisms underlying the prognostic differences between the molecular subtypes but also provide a theoretical basis for the future development of precision therapeutic strategies. We have incorporated the relevant findings into the Discussion section of the revised manuscript. Once again, we sincerely thank the reviewer for the insightful suggestions, which have greatly contributed to the improvement of our study.

Line 351-366

5�Training and validation sets: The authors mentioned splitting the dataset in two (training and validation set), but it is unclear whether this was done for the IPF dataset or mistakenly refers to a "sepsis" dataset (line 117). Please clarify and ensure consistency in terminology.

Reply

We sincerely thank the reviewer for pointing out the ambiguity in our dataset partitioning and terminology usage. In response to this issue, we have conducted a thorough review and correction of the relevant content in the revised manuscript.

Upon careful verification, we confirmed that the original use of the term “sepsis dataset” was incorrect; the accurate term should be “IPF dataset.” Specifically, we randomly divided 176 IPF patient samples with complete clinical information from the GSE70866 dataset into a training set (n = 124) and a validation set (n = 52) at a ratio of 7:3. The training set was used for constructing the prognostic model, including univariate Cox regression, LASSO regression, and multivariate Cox regression analyses. The validation set was used to assess and verify the model’s performance, thereby ensuring its robustness and generalizability.

We have updated and standardized the relevant terminology in the “Materials and Methods” section and added a detailed description of the sample partitioning criteria and workflow. These revisions ensure consistency, accuracy, and improve the reproducibility and clarity of our study. Once again, we sincerely appreciate the reviewer’s careful review and valuable suggestions.

Line 107-125

6: Sankey Diagram: The

---

## [Decision Letter · Decision Letter 1]

PONE-D-25-16281R1Comprehensive analysis of molecular characteristic and clinical prognosis of CD8+ T cell related genes in idiopathic pulmonary fibrosisPLOS ONE

Dear Dr. Liu,

Thank you for submitting your manuscript to PLOS ONE. After careful consideration, we feel that it has merit but does not fully meet PLOS ONE’s publication criteria as it currently stands. Therefore, we invite you to submit a revised version of the manuscript that addresses the points raised during the review process.

In particular, although you answered to the reviewer 1, you forgot to answered to the queries of reviewer 2. To facilitate you, I report below his/her original queries: 

Thank you for the opportunity to review this well conducted and very interesting study that further supports the Role of immunity in IPF. This study was an unmet need. While new data were available for monocytes and Tregs in IPF, there was a need for such studies for CD8 T cells. The analysis is elegant and the figures are nice ( volcano plots, Survival curves, heat maps etc all of them are elegant). Some comments that could even improve more this manuscript are provided. They mainly reflect the effort to identify therapeutic targets and to further improve the discussion based on other recent works for immunity in IPF. Specific comments follow:

1) Authors should do connectivity map analysis using as input the CD8 related genes that they found to find drugs that increase or decrease these genes. This can help the field move on. For example they can use the Clue platform for that which accepts as input up to 250 genes ( in case the genes are more, those that have the higher log2fold change are selected).

2) Authors should substantially improve their discussion. First of all, it would be better to start it with the main findings of this work and not with introductory sentences. Secondly authors need to discuss more immune abberrations in IPF and how their work is relevant compared to other recent works. Single cell RNA sequencing studies in IPF found an outcome-predictive increase in classical monocytes and T regulatory cells. Further investigation showed that CD14+CD163–HLA-DRlow monocytes were the monocytes that were most consistently associated with progression. These monocytes seem to be associated with expansion of Tregs, suppression of natural killer cells and suppression of some T cell sub populations. This could be the reason that expansion of Tregs, and suppression of natural killer cells have been associated with IPF progression.

Further research aiming to address whether CD8 T cells are part of the same cascade and for their exact interaction with other immune subpopulations is greatly anticipated.

The studies that describe the above are provided here and should be in the reference list along with a more detailed presentation of immunity in IPF in the discussion.

https://pubmed.ncbi.nlm.nih.gov/38717443/

https://www.medrxiv.org/content/10.1101/2024.08.07.24311386v2

https://pmc.ncbi.nlm.nih.gov/articles/PMC8491266/

https://pmc.ncbi.nlm.nih.gov/articles/PMC12001815/

https://pmc.ncbi.nlm.nih.gov/articles/PMC3570653/

https://www.thelancet.com/journals/ebiom/article/PIIS2352-3964(23)00332-8/fulltext

https://pubmed.ncbi.nlm.nih.gov/39510135/

3) Authors should change a title of the results by writing “Validation of the accuracy” instead of “Validate the accuracy of” so that they are consistent with the other titles of results

We look forward to receiving your revised manuscript.

Kind regards,

Simone Agostini, Ph.D.

Academic Editor

PLOS ONE

Reviewers' comments:

Reviewer's Responses to Questions

**Comments to the Author**

1. If the authors have adequately addressed your comments raised in a previous round of review and you feel that this manuscript is now acceptable for publication, you may indicate that here to bypass the “Comments to the Author” section, enter your conflict of interest statement in the “Confidential to Editor” section, and submit your "Accept" recommendation.

Reviewer #1: All comments have been addressed

Reviewer #2: (No Response)

2. Is the manuscript technically sound, and do the data support the conclusions?

Reviewer #1: Yes

Reviewer #2: Yes

3. Has the statistical analysis been performed appropriately and rigorously? 

Reviewer #1: Yes

Reviewer #2: Yes

4. Have the authors made all data underlying the findings in their manuscript fully available?

Reviewer #1: Yes

Reviewer #2: Yes

5. Is the manuscript presented in an intelligible fashion and written in standard English?

Reviewer #1: Yes

Reviewer #2: Yes

6. Review Comments to the Author

Reviewer #1: all the reviewer's comments have been perfectly addressed (including major and minor changes specified), improving the rigor and scientific credibility of the methodology, results and the implications of the study

Reviewer #2: Authors did not respond to any of my comments

Either they were not able to see them for some reason through the journal site or they did't want to.

7. PLOS authors have the option to publish the peer review history of their article (what does this mean? ). If published, this will include your full peer review and any attached files.

**Do you want your identity to be public for this peer review?** For information about this choice, including consent withdrawal, please see our Privacy Policy .

Reviewer #1: No

Reviewer #2: No

---

## [Author Response · Author response to Decision Letter 2]

27 Jun 2025

Reviewer #2: Thank you for the opportunity to review this well conducted and very interesting study that further supports the Role of immunity in IPF. This study was an unmet need. While new data were available for monocytes and Tregs in IPF, there was a need for such studies for CD8 T cells. The analysis is elegant and the figures are nice ( volcano plots, Survival curves, heat maps etc all of them are elegant). Some comments that could even improve more this manuscript are provided. They mainly reflect the effort to identify therapeutic targets and to further improve the discussion based on other recent works for immunity in IPF. Specific comments follow:

1�Authors should do connectivity map analysis using as input the CD8 related genes that they found to find drugs that increase or decrease these genes. This can help the field move on. For example they can use the Clue platform for that which accepts as input up to 250 genes ( in case the genes are more, those that have the higher log2fold change are selected).

Reply:

We sincerely thank the reviewer for this insightful and constructive suggestion. We fully agree that performing a Connectivity Map (CMap) analysis based on CD8-related genes may help identify potential compounds that modulate these targets and thereby provide valuable clues for developing immunotherapeutic strategies.

In fact, we attempted to conduct CMap analysis using the Clue platform during our exploratory analysis. However, the number of CD8-related differentially expressed genes identified in our study was relatively limited (only 48 genes), and the analysis did not yield statistically significant or biologically meaningful drug predictions. As a result, we decided not to include this part in the current manuscript in order to maintain analytical rigor.

Nevertheless, we have added a description of this exploratory attempt and its limitation in the revised Discussion section. We also pointed out that future studies, potentially based on expanded sample size or integrated immune-related signatures, could allow more robust drug prediction through CMap or similar approaches.

Once again, we appreciate your thoughtful suggestion, which has provided us with an important direction for future research.

2� Authors should substantially improve their discussion. First of all, it would be better to start it with the main findings of this work and not with introductory sentences. Secondly authors need to discuss more immune abberrations in IPF and how their work is relevant compared to other recent works. Single cell RNA sequencing studies in IPF found an outcome-predictive increase in classical monocytes and T regulatory cells. Further investigation showed that CD14+CD163–HLA-DRlow monocytes were the monocytes that were most consistently associated with progression. These monocytes seem to be associated with expansion of Tregs, suppression of natural killer cells and suppression of some T cell sub populations. This could be the reason that expansion of Tregs, and suppression of natural killer cells have been associated with IPF progression.

Further research aiming to address whether CD8 T cells are part of the same cascade and for their exact interaction with other immune subpopulations is greatly anticipated.

The studies that describe the above are provided here and should be in the reference list along with a more detailed presentation of immunity in IPF in the discussion.

https://pubmed.ncbi.nlm.nih.gov/38717443/

https://www.medrxiv.org/content/10.1101/2024.08.07.24311386v2

https://pmc.ncbi.nlm.nih.gov/articles/PMC8491266/

https://pmc.ncbi.nlm.nih.gov/articles/PMC12001815/

https://pmc.ncbi.nlm.nih.gov/articles/PMC3570653/

https://www.thelancet.com/journals/ebiom/article/PIIS2352-3964(23)00332-8/fulltext

https://pubmed.ncbi.nlm.nih.gov/39510135/

• https://pubmed.ncbi.nlm.nih.gov/38717443/

• https://www.medrxiv.org/content/10.1101/2024.08.07.24311386v2

• https://pmc.ncbi.nlm.nih.gov/articles/PMC8491266/

• https://pmc.ncbi.nlm.nih.gov/articles/PMC12001815/

• https://pmc.ncbi.nlm.nih.gov/articles/PMC3570653/

• https://www.thelancet.com/journals/ebiom/article/PIIS2352-3964(23)00332-8/fulltext

• https://pubmed.ncbi.nlm.nih.gov/39510135/

Reply

We sincerely appreciate the reviewer’s insightful comments, which have helped us significantly improve the discussion section of our manuscript. As suggested, we have restructured the discussion to begin with a clear summary of our main findings before elaborating on their implications. Additionally, we have expanded our discussion on immune aberrations in IPF, particularly the role of Tregs, and their interactions with other immune subsets such as NK cells and effector T cells. We now highlight recent single-cell RNA sequencing studies linking CD14+CD163–HLA-DRlow monocytes to disease progression and their potential role in Treg expansion and NK cell suppression, aligning our findings with this emerging evidence. We have also incorporated the recommended references into the revised manuscript, which further strengthens the scientific rigor and context of our work. We are grateful for these suggestions, as they have enhanced the depth and accuracy of our discussion. Thank you for your valuable feedback.

Line 369-Line 383

3) Authors should change a title of the results by writing “Validation of the accuracy” instead of “Validate the accuracy of” so that they are consistent with the other titles of results

Reply

We sincerely appreciate the reviewer’s careful attention to detail. As suggested, we have revised the title of the results section from "Validate the accuracy of" to "Validation of the accuracy" to maintain consistency with the other section titles. Thank you for this helpful correction.

Line 277

---

## [Decision Letter · Decision Letter 2]

Comprehensive analysis of molecular characteristic and clinical prognosis of CD8+ T cell related genes in idiopathic pulmonary fibrosis

PONE-D-25-16281R2

Dear Dr. Liu,

We’re pleased to inform you that your manuscript has been judged scientifically suitable for publication and will be formally accepted for publication once it meets all outstanding technical requirements.

Kind regards,

Simone Agostini, Ph.D.

Academic Editor

PLOS ONE

Additional Editor Comments (optional):

Reviewers' comments:

Reviewer's Responses to Questions

**Comments to the Author**

1. If the authors have adequately addressed your comments raised in a previous round of review and you feel that this manuscript is now acceptable for publication, you may indicate that here to bypass the “Comments to the Author” section, enter your conflict of interest statement in the “Confidential to Editor” section, and submit your "Accept" recommendation.

Reviewer #2: All comments have been addressed

2. Is the manuscript technically sound, and do the data support the conclusions?

Reviewer #2: Yes

3. Has the statistical analysis been performed appropriately and rigorously? 

Reviewer #2: Yes

4. Have the authors made all data underlying the findings in their manuscript fully available?

Reviewer #2: Yes

5. Is the manuscript presented in an intelligible fashion and written in standard English?

Reviewer #2: Yes

6. Review Comments to the Author

Reviewer #2: (No Response)

7. PLOS authors have the option to publish the peer review history of their article (what does this mean? ). If published, this will include your full peer review and any attached files.

**Do you want your identity to be public for this peer review?** For information about this choice, including consent withdrawal, please see our Privacy Policy .

Reviewer #2: No

---

## [Editor Report · Acceptance letter]

PONE-D-25-16281R2

PLOS ONE

Dear Dr. Liu,

I'm pleased to inform you that your manuscript has been deemed suitable for publication in PLOS ONE. Congratulations! Your manuscript is now being handed over to our production team.

Kind regards,

on behalf of

Dr. Simone Agostini

Academic Editor

PLOS ONE